# Sustained Improvement in the Management of Patients with Non-Small-Cell Lung Cancer (NSCLC) Harboring ALK Translocation: Where Are We Running?

Gianluca Spitaleri [1,*], Pamela Trillo Aliaga [1], Ilaria Attili [1], Ester Del Signore [1], Carla Corvaja [1], Chiara Corti [2,3], Edoardo Crimini [2,3], Antonio Passaro [1] and Filippo de Marinis [1]

1   Division of Thoracic Oncology, IEO, European Institute of Oncology IRCCS, Via Ripamonti 435, 20141 Milan, Italy
2   Division of New Drugs and Early Drug Development for Innovative Therapies, European Institute of Oncology, IRCCS, Via Ripamonti 435, 20141 Milan, Italy
3   Department of Oncology and Haematology (DIPO), University of Milan, 20122 Milan, Italy
*   Correspondence: gianluca.spitaleri@ieo.it

**Abstract:** ALK translocation amounts to around 3–7% of all NSCLCs. The clinical features of ALK+ NSCLC are an adenocarcinoma histology, younger age, limited smoking history, and brain metastases. The activity of chemotherapy and immunotherapy is modest in ALK+ disease. Several randomized trials have proven that ALK inhibitors (ALK-Is) have greater efficacy with respect to platinum-based chemotherapy and that second/third generation ALK-Is are better than crizotinib in terms of improvements in median progression-free survival and brain metastases management. Unfortunately, most patients develop acquired resistance to ALK-Is that is mediated by on- and off-target mechanisms. Translational and clinical research are continuing to develop new drugs and/or combinations in order to raise the bar and further improve the results attained up to now. This review summarizes first-line randomized clinical trials of several ALK-Is and the management of brain metastases with a focus on ALK-I resistance mechanisms. The last section addresses future developments and challenges.

**Keywords:** NSCLC; ALK; ALK inhibitors; alectinib; lorlatinib; brigatinib; ceritinib; ensartinib; brain metastases; resistance mechanisms

## 1. Introduction

Anaplastic lymphoma kinase (ALK) was first detected in a subset of anaplastic large-cell lymphomas in 1994 [1]. The first description of echinoderm microtubule-associated protein-like 4 (EML-4)-ALK rearrangement in non-small-cell lung cancer (NSCLC) was reported in a Japanese male former smoker in 2007 [2]. Physiologically ALK is expressed only in the brain and spinal cord of embryos, and it is essential for neurological development [3]. In adulthood, ALK is constitutively expressed in limited nervous tissues. The aberrant expression and activation of ALK fusion proteins in cells leads to cellular transformation through a signaling network which involves the activation of the JAK/STAT3, PI3K/AKT/mTOR, and RAS/ERK pathways, which are essential for cell proliferation, cycling, and survival (Figure 1) [4]. In addition, several EML4-ALK variants and new partner genes (e.g., KIF5B, HIP1, BIRC6, MYT1L, and MPRIP, among others) have been identified [5–9] (Figure 2).

ALK amounts to around 3–7% of all NSCLCs and around 12% of all lung adenocarcinomas [10–12]. Moreover, it is associated with a more aggressive histologic grade. The frequency of ALK is higher in younger patients (median age around 50 years old), in females, and in patients with a limited smoking history (never or <10 pack-years) [10–12]. The incidence of ALK seems to be the same, irrespective of ethnicity [10]. Nevertheless,

all non-squamous NSCLCs, regardless of these clinical features, should be tested for ALK, either by Fluorescence In Situ Hybridization (FISH) and/or by immunohistochemistry (IHC) [13]. It has been reported that concomitant mutations of other molecular gene drivers are extremely rare, representing less than 2% [14,15].

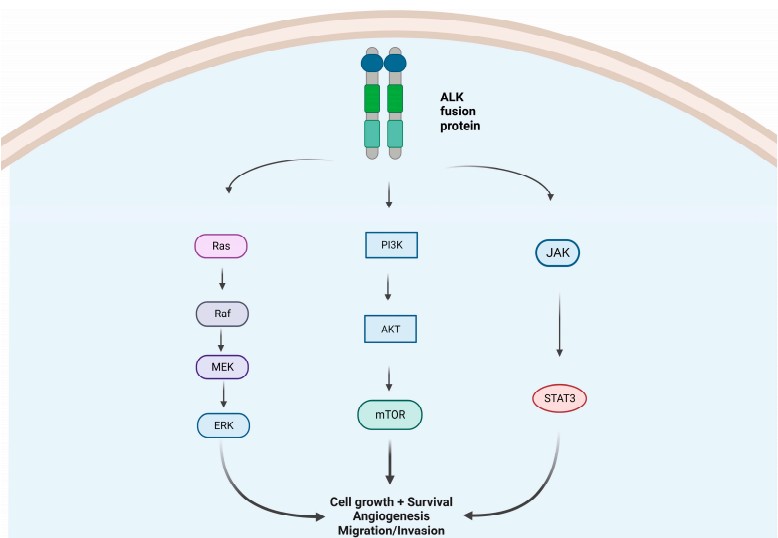

**Figure 1.** Intracellular molecular pathways due to an aberrant ALK-fusion protein oncogene: MAPK/ERK pathway, IK3/AKT/mTOR pathway, and JAK/STAT pathway. Abbreviations: Ras = Rat sarcoma virus proteins; Braf = serine/threonine-protein kinase B-Raf (v-Raf murine sarcoma viral oncogene homolog B); MEK = mitogen-activated protein kinase kinase; ERK = extracellular-signal-regulated kinases; PIK3 = phosphatidylinositol 3-kinases; AKT = protein kinase B (also called PKB); mTOR= mammalian target of rapamycin protein. JAK = janus kinase. STAT3 = signal transducer and activator of transcription 3. Created by biorender.com.

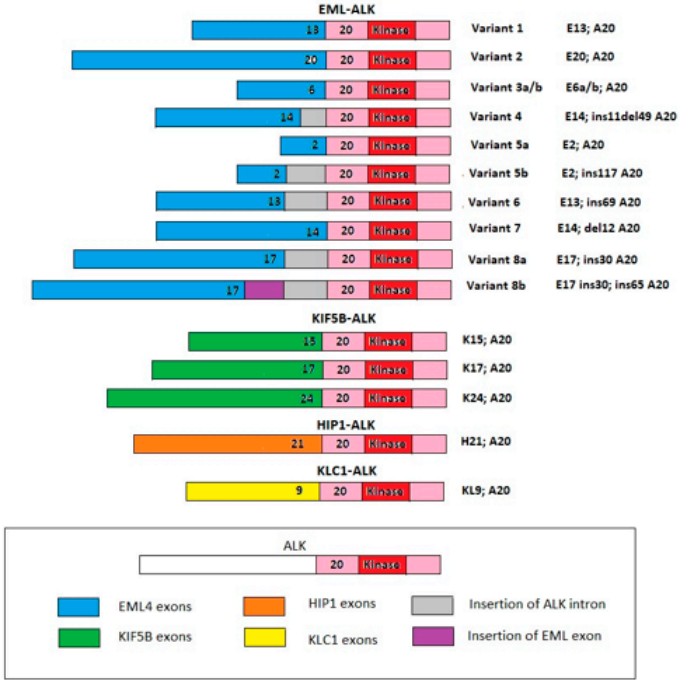

**Figure 2.** The most common ALK fusion genes reported in NSCLC. EML/ALK fusion is the most well-known. There are several variants for this fusion, variant 1 being the most common, followed by variant 3 and variant 2. There are also other fusion partners for ALK (some are drawn in the figure). Abbreviations: KIF5B = kinesin 5B family member; HIP1 = huntingtin interacting protein; KLC1 = kinesin light chain 1.

Moreover, patients with ALK+ NSCLC are more likely to receive a diagnosis of metastatic disease, and the most frequent sites of metastases are the pericardium, pleura, and liver [16]. Patients with ALK+ NSCLC are more likely to have brain metastases, particularly when they have advanced disease [17]. At the time of diagnosis of advanced disease, the incidence of brain metastases is around 25%, and their occurrence can increase to up to 45% within three years of survival with the use of non-penetrating brain barrier targeted therapies [18].

The presence of ALK seems to worsen the clinical outcomes of patients, particularly in the advanced setting [19–21]. In the pre-targeted therapy era, the 5-year OS rate for molecularly unselected stage IV NSCLC was approximately 2% [22].

The impact of chemotherapy on ALK+ NSCLC is modest, even though this disease seems to be more sensitive to pemetrexed-based regimens with respect to ALK- NSCLC [23–25], maybe due to the fact that ALK+ adenocarcinoma has the lowest levels of thymidylate synthase [25–27]. Three randomized clinical trials have confirmed the good performance of pemetrexed in ALK+ NSCLC, both in the second-line setting (PROFILE-1007) and in the first-line setting (PROFILE-1014 and ASCEND-4): in the second-line setting, the overall response rate (ORR) to pemetrexed was 29% with a median-progression-free survival (mPFS) of 4.2 months; in the first-line setting, the ORR was 27–45% with a mPFS of 7.0–8.1 months [28–30]. However, the activity of pemetrexed seems to vanish in patients who have been heavily pre-treated, as reported in the ASCEND-5 (post-crizotinib and up two lines of chemotherapy) and ALURA (post-crizotinib and one line of chemotherapy) trials, where the mPFS was 2.9 and 1.6 months, respectively [31,32].

The IMMUNOTARGET registry shows that the activity of immunotherapy (IO) in ALK+ NSCLC is poor: in 23 patients with ALK+ NSCLC, none responded to IO with mPFS of 3.1 months. Neither smoking exposure nor PD-L1 expression augmented the activity of this treatment in this subset of patients [33]. Hence, immunotherapy is not a promising treatment.

Instead, since the advent of ALK inhibitors (ALK-I), the treatment landscape and prognosis of ALK+ NSCLC patients have been radically revolutionized. Two different retrospective analyses have shown longer median survival lengths of 6.8 and 4.3 years, respectively [34,35].

Since the introduction of crizotinib, the first-in-class ALK-I, the treatment scenario has been continuously changing. This review summarizes the first-line randomized clinical trials of several ALK-Is. Special attention is paid to the management of brain metastases and the development of resistance mechanisms to ALK-Is. The last section of the review addresses future developments.

## 2. First-Line Randomized Clinical Trials of ALK-Is

PROFILE-1014 was the first randomized clinical trial of an ALK-I. It was designed to compare crizotinib versus first-line chemotherapy (cisplatin or carboplatin plus pemetrexed) in 343 patients with advanced ALK+ (as determined centrally with the use of a Vysis ALK Break Apart FISH Probe Kit) non-squamous NSCLC who had not received previous systemic treatment for advanced disease [29]. Crossover to crizotinib treatment after disease progression was permitted for patients who had previously been assigned to the control arm. The primary endpoint was PFS as assessed by independent radiologic review (BIRC-PFS). The ORR of crizotinib was 74% versus 45% of the control arm. The mPFS was statistically significantly longer for crizotinib (10.9 months) respect to chemotherapy (7 months) with a hazard ratio (HR) of 0.45 (95% CI, 0.35 to 0.60; $p < 0.001$). This benefit was observed across all the subgroups. The safety profile of crizotinib was acceptable. The most common adverse events (AEs) with crizotinib were vision disorders, diarrhea, nausea, and edema. Grade (G) 3–4 AEs were 54% and 5% resulted in permanent drug discontinuation. The 4-year overall survival (OS) rate was 56.6% (95% CI, 48.3% to 64.1%) in the arm assigned to crizotinib [36]. In the ASCEND-4 trial, 376 patients with stage IIIB/IV ALK+ (centrally tested by VENTANA anti-ALK, D5F3 IHC assay) non-squamous NSCLC were

randomized 1:1 to ceritinib (*n* = 189) or chemotherapy (*n* = 187) [30]. Patients randomized to chemotherapy were allowed to crossover to ceritinib at the disease progression. The primary endpoint was BIRC-PFS. The mPFS was 16.6 months in the ceritinib group versus 8.1 months in the chemotherapy group with a HR 0.55 (95% CI 0.42–0.73, *p* < 0.00001). Regarding toxicity: the G3/4 AEs rate was 78%, and dose reductions and discontinuation were 28% and 2%, respectively. The most common AEs related to ceritinib were diarrhea (85%), nausea (69%), vomiting (66%), and an increase in alanine aminotransferase (ALT) (60%). The performance of ceritinib was jeopardized by two findings in two scales of quality of life 'QLQ-C30 instrument': chemotherapy was more favorable than ceritinib for diarrhea and nausea/vomiting scales.

The Global ALEX trial was the first one to compare two ALK-Is in treatment-naïve patients with ALK+ (centrally tested by VENTANA anti-ALK, D5F3 IHC assay) NSCLC [37]. The investigators randomized 303 patients to receive either alectinib or crizotinib. Unlike the prior trials, the primary endpoint was investigator-assessed PFS. Cross-over was not allowed. The BIRC mPFS was significantly longer with alectinib (25.7 months) than with crizotinib (10.4 months) with a HR 0.50 (95% CI 0.36 to 0.70, *p* < 0.001). The benefit with Alectinib was consistent for all subgroups save for active smokers and patients with an ECOG PS of 2. As for the principal endpoint of the trial, the investigator assessed mPFS with alectinib was 34.8 months versus 10.9 months with crizotinib (HR 0.43, 95% CI 0.32–0.58). The OS, after a median follow-up of 48 months, was still immature with an estimated 5-year OS rate of 62.5% with alectinib and 45.5% with crizotinib [38]. The safety profile of alectinib was good: the G3-4 AEs rate was 52%, adverse events leading dose reductions and treatment discontinuation were 52% and 20%, respectively [38]. The most common AEs of alectinib were anemia (20%), myalgia (16%), increased blood bilirubin (15%), increased weight (10%), musculoskeletal pain (7%), and photosensitivity reaction (5%).

In ALTA 1L trial, 275 patients with advanced ALK + (locally tested by Ventana IHC assay and/or FISH) NSCLC, who had not previously received ALK-I, were randomized to receive brigatinib or crizotinib [39]. In the crizotinib group, crossover to brigatinib was permitted after disease progression. The primary endpoint was BIRC-PFS. After a follow-up period of 40 months, the BIRC m-PFS with brigatinib was 24 months versus 11.1 months with crizotinib (HR = 0.48, 95% CI: 0.35–0.66). The mPFS with brigatinib reached 30.8 months; the estimated 4-year OS with brigatinib was 66% [40,41]. The most common AEs were gastrointestinal events, increased blood creatine phosphokinase, cough, and increased aminotransferases. The main indicators of brigatinib toxicity were a G3/4 rate (78%) and adverse events leading to a dose reduction and treatment discontinuation (44% and 13%, respectively) [41].

In the eXalt3 trial, 290 naïve patients with ALK+ (locally tested by IHC and or FISH and after protocol amendment was centrally confirmed) NSCLCs were randomized to receive ensartinib or crizotinib [42]. The principal endpoint was BIRC-PFS, which was assessed in all randomized patients and in patients enrolled after a major protocol amendment (mITT). Crossover was not permitted. After a median follow-up of 24 months, the BIRC mPFS with ensartinib was 25.8 months versus 12.7 months with crizotinib with an HR of 0.51 (95% CI 0.35–0.72, *p* < 0.001). In the mITT population, the median PFS in the ensartinib group was not reached, and the median PFS in the crizotinib group was 12.7 months (HR 0.45; 95% CI, 0.30–0.66; *p* < 0.001). The G3-4 AE rate with ensartinib was 50%, while the rates of adverse events leading to dose reduction and treatment discontinuation were 24% and 9%, respectively. The most common events with ensartinib were skin rash, aspartate aminotransferase (AST) increase, ALT increase, pruritus, nausea, and edema [42].

In the CROWN trial, 296 naïve patients with advanced ALK+ (locally tested by FISH or IHC assay) NSCLC were randomized to receive lorlatinib or crizotinib [43]. The principal endpoint was BIRC mPFS. Crossover was not allowed. At the median follow-up of 33 months, the BIRC mPFS with lorlatinib was still not reached while the mPFS with crizotinib was 9.1 months (HR, 0.19; 95% CI, 0.131–0.274) [44]. The G3-4 toxicity rate was 75% (63% treatment-related), while the rates of adverse events leading to a dose

reduction or treatment discontinuation were 21% and 11%, respectively [44]. The most common adverse events with lorlatinib were hyperlipidemia, edema, increased weight, peripheral neuropathy, and cognitive effects. Importantly, in a post-hoc analysis, the high incidence of dyslipidemia did not translate to a higher risk of cardiovascular events [44]. Tables 1 and S1 summarize the results from global first-line clinical trials of ALK-Is and their main features, respectively.

**Table 1.** First-line clinical trials of ALK-Is.

| Trial | Number * | Drug | Control Arm | Principal Endpoint | N. pt (FISH-) | ORR | BIRC mPFS (mos) | 2 Year OS (%) | AEs G3/4 | Discontinuation Rate |
|-------|----------|------|-------------|--------------------|---------------|-----|-----------------|---------------|----------|----------------------|
| PROFILE-1014 [26] | NCT01154140 | Crizotinib | Pem-based CT | BIRC-mPFS | 343 | 74% | 10.9 | 71.5% | 54% | 5% |
| ASCEND-4 [27] | NCT01828099 | Ceritinib | Pem-based CT | BIRC-mPFS | 373 | 73% | 16.6 | 70.6% | 78% | 2% |
| ALEX [34,35] | NCT02075840 | Alectinib | Crizotinib | mPFS | 303 (39) | 83% | 25.7 | 72.5% | 52% | 14.5% |
| ALTA-1L [36–38] | NCT02737501 | Brigatinib | Crizotinib | BIRC-mPFS | 275 | 74% | 24.0 | 76% | 70% | 13% |
| EXALT-3 [39] | NCT02767804 | Ensartinib | Crizotinib | BIRC-mPFS | 290 (43) | 75% | 25.8 | 78% | 50% | 9% |
| CROWN [40,41] | NCT03052608 | Lorlatinib | Crizotinib | BIRC-mPFS | 296 | 78% | NR | 88% | 72% | 11% |

* ClinicalTrials.gov number; abbreviations: Pem = pemetrexed; CT = chemotherapy; BIRC mPFS = blinded independent radiologic review of median progression-free survival. Pts N = patient number; FISH–= Fluorescence In Situ Hybridization negative; IHC = immunohistochemistry; ORR = overall response rate; AEs = adverse events; G = grade. NR = not reached.

## 3. Management of Brain Metastases

The evaluation of the intracranial activity of ALK-I is of paramount importance. Considering that many patients are relatively young, it is important to achieve optimal control of brain disease and defer brain irradiation along with the potential side effects of this treatment. If a lesion is symptomatic due to the mass effect, neurosurgery may be the treatment of choice. However, multiple asymptomatic secondary lesions are usually present. In such cases, effective systemic treatment appears to be an optimal choice. ALK-Is differ in their ability to penetrate the blood–brain barrier (BBB) and, therefore, have differences in their antitumor potential. The new ALK-Is have been modified to obtain molecules with a high ability to penetrate into the central nervous system (CNS). This was achieved by changing the structures of individual molecules, which became substrates for P-gp [45] (Figure 3).

Tables 2 and S2 depict the distribution of patients with brain metastases (BM), those with target brain lesions (TLs), and the overall intracranial response rate (IC-ORR) according to several randomized trials.

From the trial of crizotinib (first generation) to that of lorlatinib (third generation), the percentage of patients who underwent prior brain radiotherapy before trial enrolment faded to less than 10%, while in the PROFILE-1014 study, patients with brain metastases needed to be treated with radiotherapy to be eligible to the trial [29]. Brain staging by magnetic resonance imaging (MRI) was mandatory for the following trials: ALEX, ALTA-1L, eXalt3, and the CROWN trial.

Patients with stable, treated BM were eligible for the PROFILE-1014 study [46]. Intracranial efficacy was assessed at baseline and every 6 or 12 weeks in patients with or without known BM, respectively. The intracranial time to tumor progression (IC-TTP) was measured as per the protocol, and a post-hoc analysis measured the intracranial disease control rate (IC-DCR). Out of 343 patients, 92 (23%) had BM at baseline. Of them, 47 underwent a baseline brain MRI. The IC-DCR values at 12 weeks and at 24 weeks were 70% and 40% for crizotinib and 45% and 25% for chemotherapy, respectively. A non-significant improvement in IC-TTP was observed with crizotinib in all patients with or without BM. The mPFS was superior for patients treated with crizotinib with respect to those treated with chemotherapy in all subgroups (with or without BM), with a worse prognosis for patients with brain metastases (9 months BM+ vs. 11 months BM−).

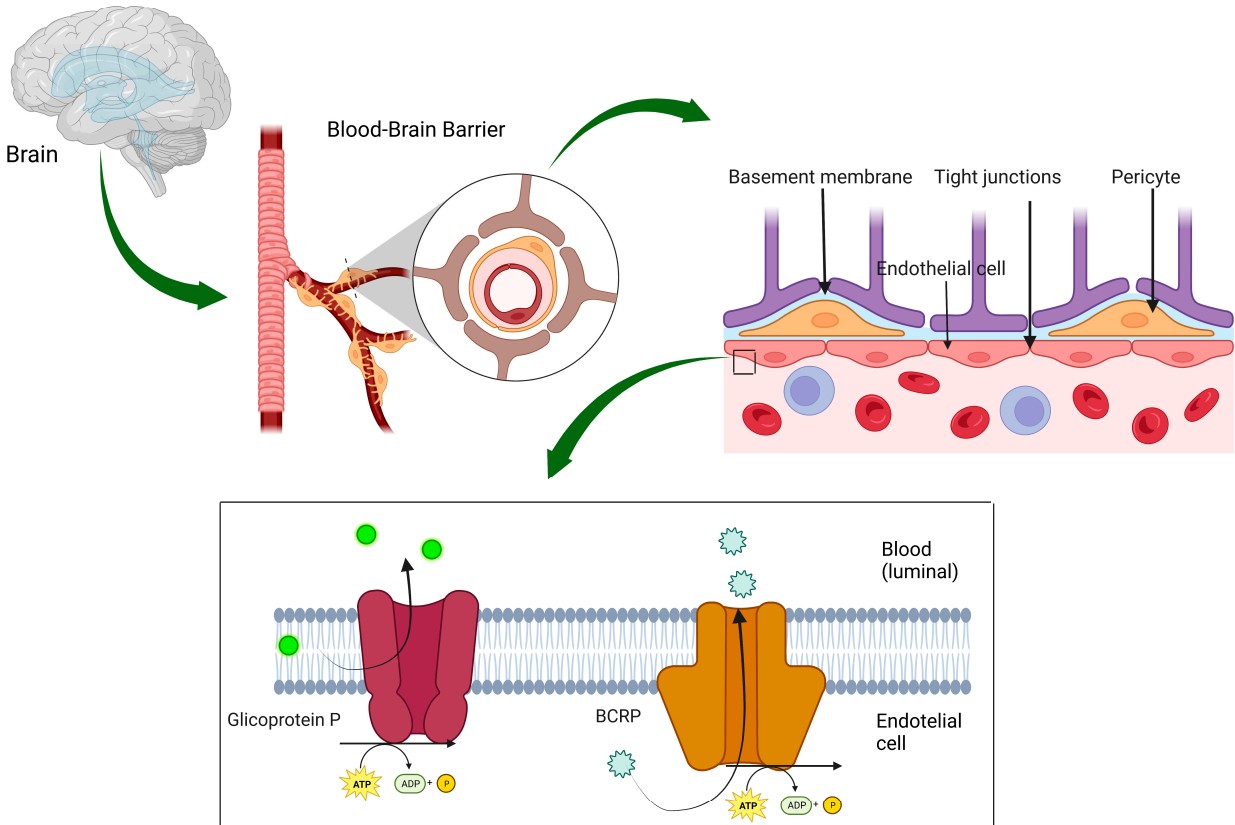

**Figure 3.** Structure of the blood–brain barrier (BBB). Tight junctions between endothelial cells in the BBB in combination with multiple transport systems, both inflow and outflow, regulate the selective movement of molecules across the BBB in the brain. In particular, many small-molecule-targeted anticancer agents (e.g., some ALK-Is) have been shown to be substrates of BBB-active efflux transporters (e.g., P-gp and BCRP), resulting in limited brain penetration by such therapies. Abbreviations: BBB = blood–brain barrier; P-gp = glycoprotein P; BCRP = breast cancer resistance protein. References: Gil M et al., Ann Med 2023 [45]. Created by biorender.com.

**Table 2.** Intracranial response (IC-ORR) in patients with brain target lesions.

| Trial | Drugs | Pts with BM | Pts with TL | Exp Arm IC-ORR | Control Arm IC-ORR |
|---|---|---|---|---|---|
| PROFILE-1014 [29] | crizotinib vs. CT | 92 | 0 | NA | NA |
| ASCEND-4 [30] | ceritinib vs. CT | 121 | 44 | 72% | 27% |
| ALEX [37,38] | alectinib vs. crizotinib | 122 | 43 | 81% | 50% |
| ALTA 1-L [39–41] | brigatinib vs. crizotinib | 96 | 41 | 78% | 26% |
| eXalt3 [42] | ensartinib vs. crizotinib | 104 | 30 | 64% | 21% |
| CROWN [43,44] | lorlatinib vs. crizotinib | 78 | 30 | 82% | 23% |

Abbreviations: Pts = patients; BM = brain metastases; TL = target lesions; IC-ORR = intracranial overall response.

In the ASCEND-4, 121 patients had baseline BM. Of them, 50 (41%) had undergone prior brain RT and 44 (36%) had brain TLs. In the 44 patients with TLs, the IC-ORRs were 72% vs. 27% for ceritinib and CT, respectively. The IC-DCR values at 12 weeks and 24 weeks were 80% vs. 75% and 70% vs. 58%, respectively, in patients with BM [30]. The efficacy of ceritinib against BM was investigated in the large non-randomized phase II trial ASCEND-7 [47,48]. One hundred and fifty-six patients with BM were enrolled in the trial and divided into 5 different arms based on their previous lines of treatment: 42 patients treated with prior brain RT and ALK-I were included in arm 1, 40 patients treated with prior ALK-I but not with brain RT were included in arm 2, 12 patients treated with prior brain RT but not with prior ALK-I were included in arm 3, 44 treatment-naïve patients were

included in arm 4, and 12 patients with leptomeningeal carcinomatosis irrespective of prior treatments were included in arm 5. Ceritinib showed modest activity against BM in the treatment-naïve patients (IC-ORR 51%) and in the patients treated with prior RT (IC-ORR 39%). In the poor prognosis group with leptomeningeal carcinomatosis, the IC-DCR was 66% with a mPFS of less than 6 months [48].

Preliminary proof of the efficacy of alectinib against BM was documented in early trials of the drug. In the United States Phase I trial, of 44 enrolled patients who progressed to treatment with crizotinib, 21 patients had BM at baseline, of whom 11 (52%) had an objective response to alectinib. Importantly, measurable concentrations of alectinib in the cerebrospinal fluid were seen in the five evaluable patients, showing the penetration of alectinib into the central nervous system [49]. In the 3-year update of the Japanese phase I trial of alectinib in 46 ALK-I-naïve NSCLC patients, the 3-year PFS rate was 62%, and 6 patients out of 14 with BM remained in the study without brain or systemic progression [50]. In the pooled analysis of the two-phase II trials of alectinib, one hundred and thirty-six patients had baseline BM (60%). Of those, 50 patients (37%) had measurable TLs at baseline. Ninety-five patients (70%) had undergone prior brain radiotherapy. For 50 patients with TLs, the IC-ORR was 64.0%. Alectinib showed activity both in patients pretreated with RT and in RT-naïve patients with IC-ORRs of 36% and 59%, respectively [51]. In the Global Alex trial, 122 patients (58 crizotinib and 64 alectinib) had BM at baseline. All patients had to undergo brain MRIs at baseline and at all time points of disease re-evaluation, as IC-mPFS was a co-primary endpoint of the trial. In patients who had been pretreated with brain RT, the IC-ORRs for alectinib and crizotinib were 86% and 71%, respectively. The difference in efficacy was even more evident in the RT-naïve patients who had IC-ORRs of 79% vs. 40%, respectively. Time to CNS progression was significantly longer with alectinib than with crizotinib and was comparable between patients with and without baseline BM ($p < 0.0001$). In the patients with BM, PFS, according to the BM status and history of radiotherapy was not reached vs. 12.7 months (pretreated with RT) and 14 vs. 7.2 months (RT-naive) after treatment with alectinib and crizotinib, respectively [52]. In a retrospective analysis of 19 patients with RT-naïve large (>1 cm) or symptomatic BM treated with alectinib, the IC-ORR was 73% with an IC median response duration of 19 months [53].

A 2-year update phase II ALTA trial confirmed durable high activity against BM by brigatinib in 222 patients with ALK+ NSCLC who progressed to crizotinib treatment randomized to two different dose regimens: arm A, 90 mg daily, and arm B, 90 mg daily for a week followed by 180 mg daily. At baseline, 71% and 67% of patients in the A and B arms had BM, respectively. The chosen regimen (arm B) showed the best control of disease in terms of the IC-ORR (56% vs. 46%), median IC-PFS (16.7 vs. 9.2 months), and OS (34 vs. 29 months) [54]. In the ALTA-1L trial, 96 patients had BM at the baseline. Of them, 41 had brain TLs. All patients had to undergo brain MRIs at baseline and at all time points of re-evaluation of disease [41]. In the 41 patients with TLs, brigatinib showed higher activity than crizotinib with an IC-ORR of 78% (vs. 26%). The median IC response duration was 28 months (vs. 9 months for crizotinib). In all 96 patients with BM, the 4-year IC PFS rate was 22% (vs. zero in patients at risk for crizotinib), and in all patients with or without BM, the 4-year rate was 46% (vs. 33% for crizotinib). Importantly, a post-hoc analysis suggested an overall survival benefit for brigatinib in patients with baseline BM (HR = 0.43, 95% CI: 0.21–0.89) [41].

A phase I/II eXalt 2 trial showed the preliminary efficacy of ensartinib against BM in the 14 patients with baseline BM TLs out of a total of 97 enrolled patients: the IC-ORR was 64.3% (with two IC complete responses and seven IC partial responses) and the IC-DCR was 93% [55]. The randomized phase III eXalt3 trial confirmed the high efficacy of ensartinib against BM compared to crizotinib [42]. In the 30 patients with TLs, the IC-ORR was 63.6% with ensartinib vs. 21.1% with crizotinib. In the patients without BM at baseline, the mPFS was NR with ensartinib vs. 16.6 months with crizotinib, and the BM- progression rates at 12 months were 4.2% and 23.9%, respectively [42].

The phase I/II trial of lorlatinib showed the first proof of efficacy of the drug against BM in both the ALK-I-naïve (IC-ORR 66%) and ALK-I-heavily-pretreated patients (IC-ORR > 50%) [56]. The CROWN trial included 78 patients (37 lorlatinib, 39 crizotinib) with BM. Of them, 30 had brain TLs. In the 30 patients with TLs, the IC-ORRs for lorlatinib and crizotinib were 82% and 23%, respectively. The HR for time to intracranial progression for lorlatinib versus crizotinib was 0.10 (95% CI 0.04–0.27); in patients without baseline BM (*n* = 112 lorlatinib; *n* = 108 crizotinib), the HR was 0.02 (95% CI 0.002–0.14). In patients without BM, one (1%) in the lorlatinib group and 25 (23%) in the crizotinib group had intracranial progression [43,44].

A retrospective trial of 90 patients with ALK+ NSCLC and BM confirmed that the administration of second generation (alectinib, brigatinib, ceritinib) or third generation (lorlatinib) ALK-I could improve the clinical outcome compared to first generation ALK-I (crizotinib) in terms of the mPFS (180 vs. 45 months), 5-year PFS rate (72 vs. 43%), and 5-year OS (76% vs. 49%) [57].

## 4. Resistance Mechanisms

When the Food and Drug Agency (FDA) approved the Vysis ALK Break Apart FISH Probe Kit as the tool to detect ALK translocation in patients with advanced NSCLC, a cutoff to meet the criteria for ALK-positive was established: ≥15% of tumor cells demonstrating a pattern of ALK probe hybridization indicative of gene rearrangement [58,59]. The 15% cut-off point was not originally defined by a clinical endpoint, but it was based on an assessment of the background signal in tissues lacking ALK gene translocation [59]. From a pooled analysis of a phase II trial (PROFILE1005) and two randomized phase III trials (PROFILE 1007 and PROFILE 1014) of crizotinib, a trend toward higher percentages was observed with a cut-off point of >15% ALK-positive cells and larger differences in ORR and PFS between crizotinib and chemotherapy [60]. In a smaller retrospective analysis by the MD Anderson Cancer Center, 66 ALK+ NSCLC patients were treated with crizotinib (56) or alectinib (10). The authors observed the best mPFS in more positive cases (if they chose a cut-off point of either 30% or 50%) [61].

As we stated earlier, the global ALEX trial enrolled patients with ALK+ NSCLC detected by VENTANA anti-ALK, D5F3 IHC assay [37]. Overall, out of 303 enrolled patients, 203 patients had FISH-positive tumors, 61 had an uninformative FISH, and 39 had ALK IHC-positive and FISH-negative tumors (21 alectinib, 18 crizotinib) [58]. In this retrospective analysis of the ALEX trial, patients with FISH-negative tumors and more selective ALK-I (alectinib) failed to be superior with respect to the multitarget agent (crizotinib) in terms of both the ORR (28% vs. 44%) and the mPFS (HR 1.33) [62].

Previously, the performance of next generation sequencing (NGS) was widespread in preclinical and translational research. Nowadays, NGS is largely applied in clinical practice, since it has several strengths: it has the ability to analyze several genes in parallel (the exact number of genes depends on the type of NGS panel), and concerning the gene fusion rearrangements, NGS helps to identify which partner gene is involved and the types of variants present [63]. Regarding ALK, at least 15 variants have been identified [64]. All variants contain the entire intracellular kinase domain of ALK, encoded by exons 20 through to 29, but differ in the point of fusion with the EML4 gene. The most common variants are variant 1 (E13;A20) (33%) and the shorter variant 3 (E6;A20) (29%) [64] (Figure 2). A preclinical study demonstrated that shorter variants (3 and 5) have less stability and are associated with less sensitivity to crizotinib [65]. A retrospective study of 54 patients with ALK+ NSCLC showed that variant 3 was associated with worse clinical outcomes when patients were treated with crizotinib or second generation ALK-Is (alectinib, ceritinib) [66]. The global ALEX and ALTA-1L trials showed that both alectinib and brigatinib were superior to crizotinib in all ALK variants; however, both trials confirmed that patients harboring ALK variant 3 had worse clinical outcomes [41,67]. In a retrospective analysis of two different cohorts of patients (129 patients from Massachusetts General Hospital plus 577 patients from Fondation in one database), Lin et al. confirmed that variant 3 was

associated with worse outcomes: the mPFS was not statistically different between variants 1 and 3 when the patients were treated with both pemetrexed-based chemotherapy and first/second generation ALK-Is, while there was a statistically difference favoring variant 3 when they were treated with lorlatinib (11 vs. 3 months). Seventy-seven patients underwent 93 re-biopsies, and a statistical difference was observed between variants 3 and 1 in terms of the percentage of resistance mechanisms (57% vs. 30%) and incidence of the most common mutation G1202R (32% vs. 0%) [68]. A subgroup analysis from the CROWN trial showed that the mPFS of lorlatinib in patients harboring variant 3 was 33 months, while those of patients receiving alectinib (from ALEX trial) and brigatinib (from the ALTA 1 L trial) were 16 and 17 months, respectively [41,44,69]. Moreover, the final analysis of ALTA-1L demonstrated that patients harboring p53+ NSCLC had worse clinical outcomes [41].

Moving on to the acquired mechanisms of resistance to ALK-Is, the mechanisms that determine insensitivity to drugs can essentially be split into two groups: the first involves the target itself via the acquisition of gene mutations or gene amplification (on-target); the second consists of heterogenous alterations that involve other molecular pathways (by-pass track or off-target) or the epithelial–mesenchymal transition, a transformation in histology, such as changing into a small-cell carcinoma (Figure 4). The acquired mutations can be localized in different ALK gene kinase domain regions: C1156Y/T, E1210K and G1269A/S are located in the kinase domain; G1202R and S1206C/Y are located in the solvent-exposed region of the kinase domain; I1171T/N/S is located in the alpha-c helix [70]. Table 3 depicts the behavior of the ALK-Is according to the most common mutations [71–83].

**Table 3.** The behavior of the ALK inhibitors according to the most common mutations [71].

| Drug | L1196M ^ | G1269A £ | C1156Y/T $ | E1210K & | I1171 α | I1151Ti β | S1206 μ | G1202R ∞ |
|---|---|---|---|---|---|---|---|---|
| Crizotinib | R | R | R | R | R | R [84] | R [84] | R [84] |
| Ceritinib | S | S | R | R | S | R | S | R |
| Alectinib | S/R | S | S | R | R | S | S | R |
| Brigatinib | S | S | S | R | S | S | R | R/S § |
| Ensartinib | S | S/R ϛ | S | R | S | S | S | R |
| Lorlatinib | S | S | S/R * | S | S | S | S | S |

Abbreviations: R = resistant; S = sensitive. References: Crizotinib ^ [72,75–77,81–86]; £ [72,75–77,81,83,85]; $ [72,76,77,85,86]; & [76]; α [76,81,85] β [76,84] μ [76,81,84] ∞ [76,77,81,84,85]; Ceritinib ^ [76]; £ [76]; $ [76] & [83]; α [71]; β [83]; μ [76]; ∞ [72,76,82,83,85,87]; Alectinib ^ R [76], S [83]; £ [76,80]; $ [76,80]; & [78] α [71,73,76,81,85]; β [80]; μ [80,82,84]; ∞ [72,76,80,82,83,85]; Brigatinib ^ [80]; £ [76,83]; $ [76,83]; & [72,76] α [73,80]; β [83]; μ [76] ∞ S [80]; R [72,76,82]; Ensartinib ^ [77]; £ S [83]; R [77]; $ [77] & [77]; α [77]; β [77]; μ [77]; ∞ [77,83]; Lorlatinib S single mutations [72,74,78,79,82,83,85]. However, these studies detected compound mutations conferring resistance to lorlatinib. Notably, a retrospective study was conducted on 17 patients who underwent biopsies after crizotinib/alectinib progression [76]. ϛ Phase II trial of ensartinib (post crizotinib) [77]. * Retrospective analysis of 15 liquid biopsies after lorlatinib [78]. § J-ALTA [80].

Several landmark retrospective analyses helped us to better understand the molecular basis of the development of resistance to TKI [76,78,81–83]. Gainor et al. analyzed 103 re-biopsies from 83 patients. Of these, 14 (18%) patients underwent re-biopsies after crizotinib and after a second generation ALK-I: 9 who received ceritinib, 3 who received alectinib, and 2 who received brigatinib. They analyzed the 55 samples attained after progression on crizotinib, and they identified that all evaluable samples for FISH were positive for ALK, and 31% had on-target alterations: 10 (20%) had crizotinib-resistant mutations (the most frequent were L1196M, G1269A, and G1202R), while 8.3% had ALK gene amplifications not associated with ALK gene mutations. Interestingly, the samples acquired after crizotinib and second generation ALK-I progression revealed higher percentages of acquired ALK gene mutations (50–71%) with an increased incidence of mutant solvent G1202R up to 21–50% (versus 2% of post-crizotinib samples) [76]. Liquid biopsies from pooled analyses of two prospective trials of alectinib (the global NP28673 and the North American NP28761, both enrolling patients progressing on crizotinib). The baseline liquid biopsies (post-crizotinib/pre-alectinib) revealed that 25% had acquired ALK gene mutations. Importantly, patients harboring ALK gene mutations had shorter mPFS (5.6 months) with respect to patients without secondary mutations (10.2 months), likely exclusively

due to BM progression over crizotinib treatment. Liquid biopsies attained after alectinib progression confirmed a higher incidence of acquired ALK gene mutations (53%): the most common were I1171 T/N/S, G1202R, and V1180L [81]. A subsequent retrospective study of 84 patients treated with a sequence of a second generation ALK-I and lorlatinib showed that the incidence of ALK mutations was similar in tissue (63%) and liquid biopsies (67%) after alectinib treatment and was even higher after progressing on lorlatinib (76%) with a detection rate of 48% (vs. 23% after alectinib) for compound mutations (≥2 ALK gene mutations) [78]. A different study of 116 patients treated with a complete sequence of first, second, and third generation ALK-Is confirmed that continuous pression on the ALK gene can accelerate the accumulation of ALK resistance mutations and may lead to treatment-refractory compound ALK mutations [82]. A post-hoc analysis of baseline plasma and tumor tissue samples collected from 198 patients with ALK+ NSCLC from the phase II study of lorlatinib showed that patients harboring ALK mutations experienced major benefits from lorlatinib with respect to patients without mutations in terms of the ORR (about 60% vs. 30%) and mPFS (7.3 vs. 5.5 months for the plasma group, 11 vs. 5.4 months for the tissue group), respectively [83].

Activation of the bypass signaling pathways accounts for around 16% of cases of resistance to ALK-Is and consists of gene alterations outside the ALK gene. Thus far, many alterations have been identified: the Epidermal Growth Factor Receptor (EGFR) pathway; c-kit amplification; IGFR1R, MAPK, and MET amplification; the BRAF gene; the YAP gene; and NF2 deletion mutations, among others [77,79,84,88–95] (Table S3). The incidence of SCLC transformation in ALK disease is very rare (0.8%, 2/263); however, the cases seem to increase moving from first to third generation ALK-Is: no cases were identified after crizotinib progression (0/95); 0.8% were identified after second generation ALK-I (1/130); and 2.6% were identified after lorlatinib treatment (2/38) [96]. The subgroup analysis of ALTA-1L from plasma samples attained from progressing patients demonstrated that these mechanisms could be even more frequent than initially assumed [41].

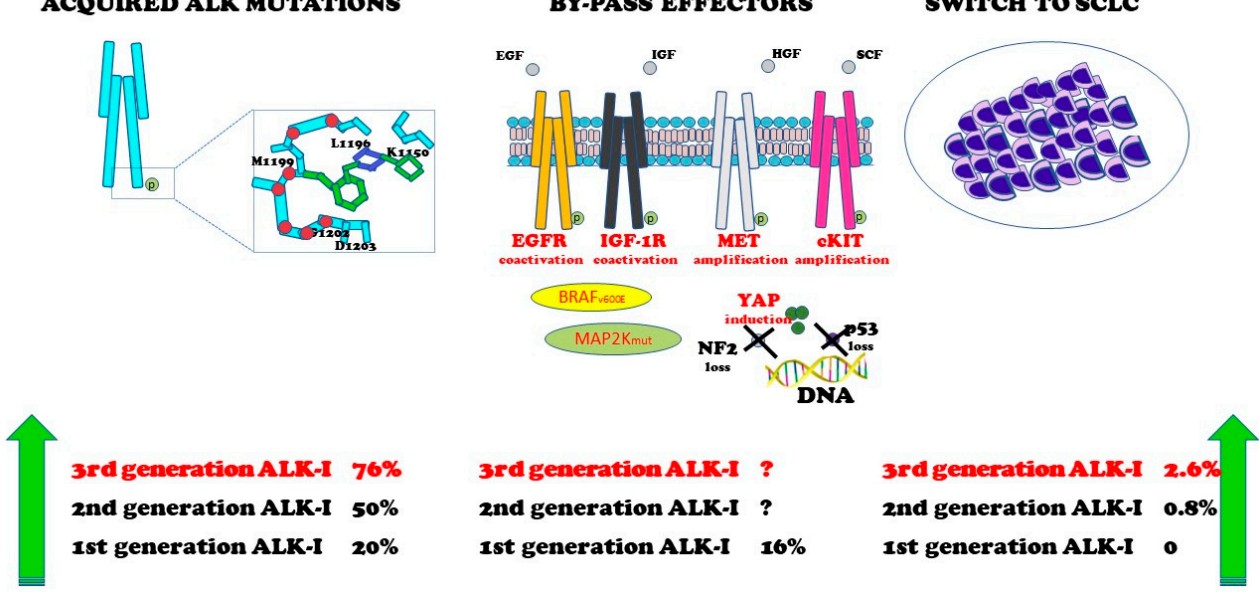

**Figure 4.** Overview of resistance mechanisms to ALK inhibitors. Abbreviations: SCLC = small-cell lung carcinoma; EGF (R) = Epidermal Growth Factor (Receptor); IGF (-1R) = Insulin Growth Factor (1 receptor); HGF = Hepatocyte Growth Factor; MET = tyrosine kinase receptor for HGF; SCF = Stem Cell Factor; cKIT = tyrosine kinase receptor for SCF; BRAF = v-raf murine sarcoma viral oncogene homolog B1; MAP2K = mitogen-activated protein kinase kinase; YAP = Yes-associated protein; NF2 = neurofibromatosis 2; DNA = deoxyribonucleic acid. References: Gainor JF et al. Cancer Discov (2016) 6 (10): 1118–1133 [76]; Shaw AT et. J Clin Oncol 2019; 37(16):1370–1379 [83]. Lin JJ et al. NPJ Precision Oncology 2020;4 (21): 1–8 [96].

## 5. Future Developments

### 5.1. Activity of ALK-I in Blood-Assessed ALK

Research on the management of ALK+ disease has been conducted in both clinical practice and in clinical research through the detection of ALK gene rearrangements via tissue-based analyses. However, it is not always possible to perform tissue biopsies. Difficulties associated with acquiring adequate tumor samples in terms of quality and quantity, spatial and temporal heterogeneity, and the repetition of tumor biopsies to assess the mechanism of resistance are the principal limitations of invasive tissue-based molecular analyses [97]. The detection of these molecular abnormalities in the plasma, called "liquid biopsies", is a valuable non-invasive complementary approach for these patients. LIQUIK (NCT04703153) and LIBL (Liquid Biopsies in Patients Presenting Non-Small-Cell Lung Cancer, NCT02511288) are two observational studies that are validating the NGS blood-assessed biomarker in 200 and 900 patients with newly diagnosed metastatic NSCLC, respectively. BFAST is a phase 2/3, global, multicenter, multicohort study designed to evaluate the safety and efficacy of targeted therapies or immunotherapies as single agents or in combination in 1000 patients with advanced NSCLC determined to harbor oncogenic somatic mutations or positive for the tumor mutational burden assay, as identified by liquid biopsies. From 2219 patients screened, 119 patients (5.4%) had ALK+ NSCLC. Of these, 87 received alectinib. The preliminary results show that, at a median follow-up of 12.6 months, there was consistent efficacy with tissue-detected ALK data attained from the other trials with an ORR of 92% and a 12-month investigator-assessed PFS of 78.4% [98]. Table S4 presents the results of these and other trials on this matter [99].

### 5.2. New 2nd/3rd ALK-Is

Table 4 reports the new second (alkotinib) and third generation (TGRX-326, foritinib, SY-3505, XZP-3621) ALK-Is that are in different phases of research [99]. Of these, the most advanced in terms of development is XZP-3621. In fact, a phase III study is underway to compare this drug to crizotinib in patients with ALK+ naïve NSCLC (NCT05204628).

**Table 4.** Ongoing trials in patients with ALK+ NSCLC progressing on ALK-Is: novel 2nd/3rd ALK-Is.

| Trial | Phase | Pt N | Drug | Class | Setting | Principal Endpoint |
|---|---|---|---|---|---|---|
| NCT03607188 | I | 18 | Alkotinib | 2nd ALK-I | Post-crizotinib | Safety |
| NCT04211922 | II | 104 | Alkotinib | 2nd ALK-I | Post-crizotinib | activity |
| NCT05441956 | I | 100 | TGRX-326 | 3rd ALK-I | Post 1st/2nd ALK-I | RP2D |
| NCT04237805 | I/II | 280 | Foritinib | 3rd ALK-I | Refractory | DLT/ORR |
| NCT05257512 | I/II | 70 | SY-3505 | 3rd ALK-I | Refractory | RP2D |
| NCT05055232 | I | 120 | XZP-3621 | 3rd ALK-I | Naïve/refractory | Safety |
| NCT05482087 | II | 190 | XZP-3621 | 3rd ALK-I | Naïve/refractory | ORR |
| NCT05204628 | RPIII | 238 | XZP-3621 | 3rd ALK-I | Naive | PFS |

Abbreviations: Pt N = patient number; RP2D = recommended phase 2 dose; DLT = dose-limiting toxicity; ORR = overall response rate; PFS = progression-free survival.

### 5.3. Overcoming Resistance to ALK-Is

The first phase of clinical research focused on the behavior of already approved ALK-Is (brigatinib, ensartinib) with respect to patients progressing on 2nd generation ALK-Is (alectinib or ceritinib). ALTA-2 is a single-arm, phase 2 trial of brigatinib involving 103 patients with advanced ALK+ NSCLC whose disease progressed on alectinib or ceritinib. The preliminary findings show that at a median follow-up of 10.8 months, there was modest activity of brigatinib in these patients: the ORR was 26.2% (29% post alectinib), the median response duration was 6.3 months, and the mPFS was 3.8 months. Resistance mutations were present in 33% (54% G1202R) of the 78 evaluable patients [100].

The second phase is dedicated to better understanding the activity of lorlatinib and the mechanism of resistance to ALK-Is. Three different trials have been designed to define the activity of lorlatinib according to prior first generation or second generation ALK-Is (ORAKLE, NCT04111705), its activity against leptomeningeal carcinomatosis

(NCT02927340), and its activity according to the ALK mutational profile defined through repeated liquid biopsies conducted during treatment (ALKALINE, NCT02927340).

The main aspect of clinical research is overcoming the mechanism of resistance. Several trials are investigating the safety and feasibility of different combinations with or without ALK-I using several classes of drug: chemotherapy, antiangiogenesis (bevacizumab), small molecule multi-TKI (lenvatinib, crizotinib, apatinib) or selective inhibitors (anti-MEK, anti-MET, or anti-SHP-2), immune checkpoint inhibitors (pembrolizumab, atezolizumab, camrelizumab, IBI-322, IBI-318), and novel immunomodulators (autologous tumor-infiltrating lymphocytes, dendritic cell vaccines, anti-CD20, natural killer cells) (Table 5) [99].

**Table 5.** Ongoing trials in patients with patients with ALK+ NSCLC progressing on ALK-Is: overcoming resistance to ALK-Is.

| Type of Therapy | Trial Identifier | Ph. | Pt N | Drug(s) | Setting | Principal Endpoint | CNS metastasis Permitted |
|---|---|---|---|---|---|---|---|
| **ALK-I** | ALTA-2 NCT03535740 | II | 103 | Brigatinib | Refractory | ORR | Asymptomatic BM; stable symptoms. |
| | ERSGATR NCT05178511 | II | 40 | Ensartinib | Refractory | ORR | Asymptomatic BM; stable symptoms; previously treated with RT. |
| | NCT02927340 | II | 30 | Lorlatinib | Naïve/ refractory | IC-DCR | LM or CM ° |
| | ORAKLE NCT04111705 | II | 112 | Lorlatinib | Refractory | ORR | Asymptomatic BM; stable symptoms; previously treated with RT. |
| | ALKALINE NCT04127110 | II | 100 | Lorlatinib | Refractory * | PFS | Asymptomatic BM/LM; stable symptoms; previously treated with RT. |
| | NCT03917043 | I | 150 | APG-2449 (multi-TKI, 3rd ALK-I) | Refractory/naïve | MDT/RP2D | BM with controlled symptoms. |
| | ALKOVE-1 NCT05384626 | I/II | 214 | NVL-655 (4th ALK-I) | Refractory # | RP2D/ORR | NR |
| **ALK-I based combo** | MASTER ALK NCT05200481 | RPII | 110 | Brigatinib +/− CT (platinum/pemetrexed) | Naïve | PFS | Symptomatic and neurologically stable BM metastases $. |
| | NCT05491811 | II | 77 | Ensartinib + bevacizumab | Naïve/p53+ | 1-yr PFS rate | CNS metastases treated with RT and/or surgery. |
| | NCT03202940 | IB/II | 31 | Alectinib + cobimetinib | Refractory & | MTD | Asymptomatic BM or treated with RT. |
| **Combo without ALK-Is** | NCT04356118 | IV | 30 | RH endostatin+ IT MTX + ALK-I | Naive/ refractory | OS % | LM only. |
| | NCT05266846 | II | 30 | CT+ bevacizumab + Pembrolizumab | Refractory | PFS | Treated CNS metastases. |
| | NCT03991403 | RPIII | 228 | CT + bevacizumab + atezolizumab | Refractory | PFS | Asymptomatic or treated BM. |
| | GFPC 06-2018 NCT04042558 | II | 149 | CT + atezolizumab+/− bevacizumab | Refractory | ORR | Asymptomatic BM. £ |
| | NCT04425135 | II | 59 | CT+ camrelizumab + apatinib | Refractory | ORR | NA |
| | NCT04989322 | II | 46 | CT+ Pembrolizumab + Lenvatinib | Refractory | ORR | Treated BM. |
| | HARMONIC NCT05456256 | II | 90 °° | CT+ LP-300 ** | Refractory | PFS/OS | Stable CNS Metastases. |
| | NCT05296278 | II | 80 | IBI-322 ## + Lenvatinib + CT | Refractory | PFS | Stable BM. |
| | NCT04777084 | Ib | 100 | IBI-318 $$ + Lenvatinib | Refractory | ORR | Asymptomatic or stable BM. |
| | NCT05681780 | I/II | 20 | CD40L-Augmented TIL + nivolumab | Refractory | AEs | Stable or treated BM. |
| | PIKACHU NCT04322890 | Obs. | 100 | Anti-PD1 plus CT | Refractory | PFS | NA |
| | NCT05195619 | I | 16 | DC vaccines + low dose CP | Refractory | AEs | NA |
| | SNK_ASTER NCT04872634 | I/II | 24 | SNK01 NK + CT +/−cetuximab | Refractory | MTD | Treated and stable CNS metastases. |
| | NCT04880863 | II | 35 | NAP+ Docetaxel+ Obinutuzumab | Refractory | ORR | Treated and stable BM. |
| | NCT03645928 | II | 178 | Autologous TIL (LN-145) + IO | Refractory | ORR, Safety | Asymptomatic and treated BM. |

**Abbreviations:** Pt N = patient number; ALK-I(s) = ALK inhibitor(s); CNS = central nervous system; BM = brain metastases; CT = chemotherapy; LM = leptomeningeal disease; CM = carcinomatous meningitis; TIL = tumor-infiltrating lymphocytes; IT = intratecal; DLT = dose-limiting toxicity; ORR = overall response rate; PFS = progression-free survival; OS = overall survival; IC-CDR = intracranial disease control rate; 1-yr = 1 year; MTD = maximum tolerated dose; AEs = adverse events; DC = autologous dendritic cell vaccine loaded with personalized peptides (PEP-DC vaccine); RH = recombinant human; NK = natural killer cells; NAP = naptumomab estafenatox, an engineered antibody–superantigen fusion protein. NA = not available. TKI = tyrosine kinase inhibitor. CP = cyclophosphamide. IO = immunotherapy; CSF = cerebrospinal fluid. % OS = calculated from leptomeningeal metastasis diagnosis; * ALK detected in tissue and blood. # Activity against compound mutation; ° presence of at least one CNS lesion (>5 mm); $ excluded LM; & in progression to alectinib monotherapy; £ included only supratentorial and cerebellar metastases. ** LP-300 is a cysteine-modifying agent; ## IBI-322 is a bispecific antibody anti-CD47/PDL1; $$ IBI-318 is a bispecific anti-PD1/PDL1. °° 90 never smoker.

Higher expectations come from novel generations of ALK-Is. APG-2449 is a novel, orally active FAK inhibitor and a third generation ALK/ROS1 TKI that has shown potent activity against a range of ALK-resistant mutations, including G1202R, L1196M, V1180L, E1210K, S1206F, G1269A, F1174L, I1171S, and C1156Y in pre-clinical NSCLC tumor models. An ongoing phase I trial is evaluating patients with second generation TKI-resistant ALK/ROS1+ NSCLC. At the time of the first trial report, 14 patients with ALK+ NSCLC had been enrolled, and the preliminary findings showed that 4/14 (28.5%) of the refractory patients had attained a PR and a patient harboring the G1202R mutation had attained a minimal response (<30%). There was an ORR of 80% among 8 naïve ALK patients. Moreover, cerebrospinal fluid PK analyses showed that APG-2449 was a brain penetrant, and among 8 patients with BM, there was 1 IC complete response (CR) and 3 IC-PR [101].

TPX-0131 and NVL-655 are the 4th "double mutant active" ALK-Is. Their strengths include a lower IC50 with respect to lorlatinib for the wild-type ALK, a comparable (if not more potent) IC50 compared with variant 1 ALK, high brain barrier penetrance, and most importantly, activity against the most common single and compound ALK mutations [102–104]. However, the phase I/II Forge-1 of TPX-0131 (NCT04849273) has prematurely closed its enrolment due to its narrow therapeutic index [105]. A phase I/II trial of NVL-655 is ongoing: phase I will determine the MTD of the drug in patients with ALK+ solid tumors. In Phase 2, study patients will be split into 4 distinct cohorts according to prior lines of ALK-Is. Table 5 summarizes these trials [99].

*5.4. Early Stage*

The Leader trial (the Lung Cancer Mutation Consortium) has developed a collaborative protocol to screen patients suffering from stage IA2-IIIA NSCLC through tissue- or blood-NGS for assessing the incidence of 10 molecular gene drivers, including ALK, in early-stage disease (NCT04712877).

In ALK+ disease, none of the TKIs have been approved for early-stage disease. Several trials are ongoing in this setting. Table S5 summarizes the ongoing trials split into neoadjuvant, adjuvant, and after chemoradiotherapy for locally advanced disease [99].

In the neo-adjuvant setting, four trials have been designed to investigate the efficacy of several ALK-Is (alectinib, brigatinib, and ensartinib) given for up to 2 cycles before surgery. The principal endpoint is the major pathological response (MPR). If a response occurs, patients are allowed to continue with adjuvant treatment.

Similar to the ADAURA trial for EGFR + NSCLC, for patients who have undergone radical surgery for IB (T > 4 cm)-IIIA (7th TNM) ALK+ NSCLC, three randomized phase III trials, ALCHEMIST, ALINA, and NCT05341583, have been carried out to rule out the efficacy of crizotinib, alectinib, amd ensartinib, respectively [106]. NCT05170204 is the largest trial for unresectable locally advanced ALK+ NSCLC and was designed to compare alectinib given for up to 3 years and durvalumab given for 1 year after chemoradiotherapy.

Both the NCCN and ESMO guidelines do not recommend any ALK-I in the early-stage setting, since we are waiting for the results from the trials of ALK-Is designed and carried out in the peri-operative stage. The introduction of ALK-Is in this setting will depend on the results attained and if they maintain the high standard derived from what they have produced in the metastatic setting. Clearly the follow-up of these trials must be longer to observe the effects on disease-free survival and, above all, overall survival, which should be the principal endpoints to look at, since the early surrogate endpoints, such as major pathologic response or complete pathological response in the neo-adjuvant trials are still not validated endpoints.

## 6. Discussion

Since its discovery in 2007, significant progress has been made in the management and prognosis of patients with ALK+ NSCLC. The overall survival of these patients treated with chemotherapy, crizotinib, and with another ALK-I (ceritinib, alectinib, or brigatinib) has reached 7.5 years, not considering the potential benefit of adding lorlatinib [34].

Crizotinib could reduce the risk of progression or death by 55% with respect to chemotherapy [29]. The gain is even higher when moving from second generation ALK-Is to lorlatinib, whereby the reduction in the risk of progression or death increases from around 50% to more than 70% with respect to crizotinib [37–44].

From a series of translational research works, we have learned that the more potent and selective the ALK-I, the longer the disease can be controlled. On the other hand, the more potent the ALK-I is, the higher the probability of generating acquired ALK mutations is: from 20% post crizotinib to more than 70% post next-generation ALK-I, with up to 50% of compound mutations occurring after lorlatinib treatment. These findings were derived from sequential treatment starting with crizotinib and ending with lorlatinib. However, lorlatinib is now available as a first-line treatment, since it has been approved by the FDA (31 March 2021) and EMA (28 January 2022). Both the NCCN and ESMO guidelines include alectinib, brigatinib, and lorlatinib as the preferred options for first-line treatment thanks to the great results attained in their pivotal randomized trials according to the long benefits on mPFS and optimal management of brain metastases, which allow us to postpone brain radiotherapy in patients with asymptomatic and/or non-large brain metastases [13,107]. At disease progresses, both international guidelines recommend going on with the same ALK-I +/− local therapy in case of oligo-progression or, in the case of systemic progression, biopsy repetition (recommendation but not mandatory). The subsequent ALK-I should be chosen on the basis of the acquired resistance mechanism. The optimal choice of upfront ALK-I and the subsequent sequence of other ALK-Is is not known due to the absence of prospective trials designed to investigate the optimal sequence of ALK-Is that could guarantee the best OS, which should be our goal, while all the pivotal randomized trials have used the mPFS as their principal endpoint. Therefore, currently, the optimal choice for the upfront ALK-I should be guided based on the available clinical data, patients' features (e.g., with or without BM at baseline), and the safety profiles of the different ALK-Is, which can affect the quality of life of the patients themselves.

However, if we glance at the strengths of lorlatinib, it is hard not to decide to start with lorlatinib. With a median follow-up time of 3 years, the mPFS for lorlatinib was still not reached, the 3-year PFS rate was 63.5%, the IC time to progression of HR for patients with baseline BM was 0.10, and most importantly, the IC progression rate for patients without baseline BM was still 1% at the 3-year follow up [108,109].

With these encouraging results, clinicians can overcome their initial resistance related to lorlatinib-related side effects, such as lipid metabolism alterations and neurocognitive effects. A post-hoc analysis from the CROWN study assures us that increases in cholesterol and triglyceride levels were not correlated with an increase in cardiovascular events [42]. After a longer follow-up, neurocognitive adverse events were less frequent than expected and seemed to be associated with known risk factors (BM, brain radiation, psychiatric illness, and use of neurotropic medications) [109,110].

Assuming that lorlatinib is the current/future gold standard for first-line ALK+ NSCLC, all subsequent lines will vanish, and therefore, we are likely to go back to the drawing board. Indeed, data from the CROWN trial suggest that 2nd ALK-Is (mostly alectinib) have modest activity: the ORR was less than 30% and that of chemotherapy was even worse, being around 17% [111]. Therefore, we may soon reach a plateau (the mPFS of lorlatinib is estimated to reach 60 months), after which the gain will be difficult to improve further. All around the world, there are many ongoing clinical trials testing different combinations with or without ALK-Is using several drug classes (chemotherapy, immune checkpoint inhibitors, antiangiogenic drugs, and even novel immune modulators), but we think little or no further improvement will occur, since they are aimed at an unselected patient population and are trying, in case of immunotherapies, to transform a cold to a hot tumor, such as ALK+ disease [112].

A different scenario could be realized if the new 4th ALK-Is (mainly NVL-655) meet their high expectations. Pre-clinical data on these drugs supersedes that of lorlatinib. They are able to inhibit several ALK mutations (including the compound mutations) at a lower

IC50 and have high brain barrier penetrance. A phase I/II trial of NVL-655 is currently ongoing (NCT05384626), and we hope that the drug can be safely and effectively administered.

## 7. Conclusions

Over the past 15 years, we have witnessed a sustained improvement in the management of patients with ALK+ NSCLC. The release of novel-generation ALK-Is has improved the clinical outcomes of these patients, attaining unexpected and dramatic results. However, most patients develop resistance to ALK-Is mediated by on- and off-target mechanisms. We are conscious that with the introduction of upfront lorlatinib, we are approaching a plateau. The activity of chemotherapy is modest, especially in patients with intracranial progression, and the role of immunotherapy seems to be very limited for this disease. Translational and clinical research is continuing to develop new drugs and/or combinations in order to raise the bar and further improve the results attained up until now.

**Supplementary Materials:** The following supporting information can be downloaded at: https://www.mdpi.com/article/10.3390/curroncol30050384/s1, Table S1: Main features of the first-line clinical trials of ALK-Is, Table S2: patients with baseline brain metastases according the several randomized trials, Table S3: Bypass signaling pathways associated with resistance to ALK inhibitors, Table S4: Ongoing trials of blood-assessed NGS to detect ALK+ disease, Table S5: summary of ongoing clinical trials of ALK-Is in the early stage disease.

**Author Contributions:** Conceptualization: G.S. and A.P.; methodology: G.S. and P.T.A.; validation: G.S. and F.d.M.; literature research G.S., P.T.A., I.A., E.D.S., C.C. (Carla Corvaja), C.C. (Chiara Corti), E.C., A.P. and F.d.M.; writing review G.S. and P.T.A., editing and approval the manuscript: G.S., P.T.A., I.A., E.D.S., C.C. (Carla Corvaja), C.C. (Chiara Corti), E.C., A.P. and F.d.M. All authors have read and agreed to the published version of the manuscript.

**Funding:** This work was partially supported by the Italian Ministry of Health with Ricerca Corrente and 5 × 1000 funds.

**Institutional Review Board Statement:** Not applicable.

**Informed Consent Statement:** Not applicable.

**Data Availability Statement:** Not applicable.

**Acknowledgments:** We thank Lenny Kravitz for inspiring the title of this review. We thank Maxim Haskins for his help with the English.

**Conflicts of Interest:** G.S. served as an advisor for Takeda outside the submitted work. A.P. served as a consultant or advisor for AstraZeneca, Boehringer Ingelheim, Bristol Myers Squibb, Eli Lilly, Janssen, Merck Sharp & Dohme, Novartis, Pfizer, and Roche/Genentech; received a speaker bureau from AstraZeneca, Boehringer Ingelheim, Daiichi Sankyo, Jansenn, Eli Lilly, Merck Sharp & Dohme, eCancer and Medscape, all outside the submitted work. F.d.M. received advisory fees from Roche, Bristol-Myers Squibb, and AstraZeneca and consulting fees from Merck Sharp & Dohme, all outside the submitted work. The other authors do not report potential conflicts of interest.

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
