# Peer review of "Sustained Improvement in the Management of Patients with Non-Small-Cell Lung Cancer (NSCLC) Harboring ALK Translocation: Where Are We Running?"

_curroncol, doi:10.3390/curroncol30050384_

Round 1

Reviewer 1 Report

The work shows the path we are taking in understanding the molecular mechanisms of lung cancer and modern thinking about the method of treatment. This is the effort of science centers all over the world. Current activities meet the criteria of specific, targeted immunotherapy. Papers from more than 40 years ago, reported improved survival in lung cancer patients with postoperative pleural empyema. Long-term immunization of the empyema was part of non-specific immunotherapy, perhaps by affecting the activity of the immune response gene (CTLA4), the gene for programmed cell death (apoptosis) on the surface of T lymphocytes (PD-1), among others. In this context, the current state of knowledge is promising for discovering new approaches for targeted molecular therapy. This paper reflects the current state of knowledge for Anaplastic Lymphoma kinase (ALK) and also shows generational changes of ALK inhibitors, which is the most important message of the study. The study is valuable, however, the "conclusions" section sounds rather literary and should more directly relate to the scientific data presented. The phrasing of the title is confusing and few sections of the paper require minor grammatical corrections. (529, 555, section 556-560, 562)

Author Response

The work shows the path we are taking in understanding the molecular mechanisms of lung cancer and modern thinking about the method of treatment. This is the effort of science centers all over the world. Current activities meet the criteria of specific, targeted immunotherapy. Papers from more than 40 years ago, reported improved survival in lung cancer patients with postoperative pleural empyema. Long-term immunization of the empyema was part of non-specific immunotherapy, perhaps by affecting the activity of the immune response gene (CTLA4), the gene for programmed cell death (apoptosis) on the surface of T lymphocytes (PD-1), among others. In this context, the current state of knowledge is promising for discovering new approaches for targeted molecular therapy. This paper reflects the current state of knowledge for Anaplastic Lymphoma kinase (ALK) and also shows generational changes of ALK inhibitors, which is the most important message of the study. The study is valuable, however, the "conclusions" section sounds rather literary and should more directly relate to the scientific data presented. The phrasing of the title is confusing, and few sections of the paper require minor grammatical corrections. (529, 555, section 556-560, 562)

Minor queries:

Q1: The study is valuable, however, the "conclusions" section sounds rather literary and should more directly relate to the scientific data presented

A1: We thank the reviewer for this observation. We have reformulated the conclusions as requested. Now we think that this section has been improved thanks to the reviewer’s suggestions.

In these 15 years, we have witnessed a sustained improvement on the management of patients with ALK+ NSCLC. The release of novel generation of ALK-Is has improved the clinical outcome of these patients reaching unexpected and dramatic results. However, most of the patients develop resistance to ALK-Is mediated by on- and off-target mechanisms. We are conscious that with the introduction of upfront lorlatinib we are approaching a plateau. The activity of the chemotherapy is modest especially in patients with intracranial progression and the role of immunotherapy seems to be very limited for this disease. Translational and clinical research are continuing to develop new drugs and/or combinations in order to raise the bar and improve further the great results attained up to now.

Q2: The phrasing of the title is confusing

A2: We have reformulated the title of the review as requested.

“Sustained improvement on the managing of patients with Non-Small-Cell Lung Cancer (NSCLC) harbouring ALK translocation: ‘where are we running?’.

Q3: few sections of the paper require minor grammatical corrections. (529, 555, section 556-560, 562)

A3: We thank the reviewer for these observations, now the English has been improved.

Reviewer 2 Report

In this review, Spitaleri G and colleagues resumed the current situation regarding the treatment landscape and prognosis in ALK positive NSCLC patients with particular emphasis on first-line randomized clinical trials of several ALK inhibitors (ALK-Is) and the management of brain metastases with a focus on ALK-Is resistance mechanisms. The authors accurately discussed the differential responses of several ALK translocations to these classes of inhibitors and proposed novel therapeutical strategies to better guide the application of ALK-TKIs in ALK translocations non-small cell lung cancer (NSCLC) patients’.

In my eyes, the paper is good considering the large number of studies proposed, ongoing clinical trials that investigate the activity of ALK inhibitors that positively affect the prognosis of ALK+ NSCLC patients. The scientific content seems good as well as the English style and language used in the manuscript need. Moreover, I appreciate the scientific efforts to organize this paper and I think that the rationale is well-discussed. The resolution of the table of each section and the reference should be improved in order to meet the quality requirements of the journal. Moreover, each paragraph was clearly discussed and corroborated with what is exposed. I didn’t observe any remarkable incongruences throughout the text.

In my opinion, this article should be considered with minor revisions in light of comments and suggestions as indicated below.

1-      I would like to suggest the authors reformulate the abstract section in virtue of the fact that the key point of the review is focused on first-line randomized clinical trials of ALK-Is, mechanisms of resistance and future challenges in ALK+ NSCLC patients. From my point of view, it didn’t emerge at first glance the clinical significance of the review and I think that it would be more appropriate to highlight these clinical and pharmacological aspects. Please proofread this.

2-      The introduction paragraph should be more deeply investigated how molecular and biological characteristics of ALK rearrangements impact intracellular signaling pathways, briefly describing the role of involved partner genes and their deregulation effects. Please discuss it.

3-      I suggest enlarging figure n.1 and improving its resolution. As it stands, it doesn’t seem sharp in line with the standard of the journal. The same should be applied to figure 2 with the difference that remains to specify the color of each ALK domain in the figure caption. Please fix it.

4-      In the context of first-line randomized clinical trials of ALK-Is, I suppose that it is essential starting with an overview of first-line clinical trials and then list the cited studies. Please move the “Table 1 summarizes the global first-line clinical trials of ALK-Is” at the end of paragraph and insert the table there. I think that it's easier to follow and flow and it's less confusing. Please fix these suggestions.

5-      As described in the point 3, please fix the figure n.3 for a better resolution and indicate each reference in the caption. In the Table 4, the references are missing. Please report them (if available) for each study.

6-      As well as for Figure 4 about the overview of resistance mechanisms to ALK inhibitors it should be improved both resolution and size.

7-       There are some mistakes in formatting table 5A and table 5B (correct form, not 5-A and 5-B). In particular, 5-B did not follow the guidelines of journal. Please adapt it in the same way as the previous ones.

8-      Into the discussion section, I would like it if it motivates the choice of the optimal upfront ALK TKI for the first-line treatment of ALK+ NSCLC, the subsequent sequencing strategies, and whether these considerations should be based on specific on-target ALK resistance or not. And if indeed the goal of treating advanced ALK+ NSCLC should not just be limited to improve median PFS in the first-line setting. Please spend a few sentences on this point.

9-      In the subparagraph about early-stage,  we should need data to definitively answer the question of the utility of ALK-directed therapies in this setting. But I think that some considerations could be addressed in this context about the potential and limitations. Please add these suggestions to the discussion section.

1-   It is ascertained that after the failure of targeted therapies, chemotherapy remains a valid option, while the role of immunotherapy is yet to be clarified. Overcoming the challenges for the development of more potent drugs will be essential to improving the survival rate of ALK+ NSCLC in the future. What scenario could be expected? Please discuss rapidly in the conclusion.

1-   Please recheck the reference list (they should be reported in this way: [1,2,3 etc] and not as indicated [1][2] throughout the text. Conflicts of Interest report some typing errors. Please correct this and fix some abbreviations and punctuation.

1-   A good paper has to come from something that's inspired as well as a good song. So I agree with your acknowledgments to Lenny Kravitz, an incredible rockstar!

Author Response

In this review, Spitaleri G and colleagues resumed the current situation regarding the treatment landscape and prognosis in ALK positive NSCLC patients with particular emphasis on first-line randomized clinical trials of several ALK inhibitors (ALK-Is) and the management of brain metastases with a focus on ALK-Is resistance mechanisms. The authors accurately discussed the differential responses of several ALK translocations to these classes of inhibitors and proposed novel therapeutical strategies to better guide the application of ALK-TKIs in ALK translocations non-small cell lung cancer (NSCLC) patients’.

In my eyes, the paper is good considering the large number of studies proposed, ongoing clinical trials that investigate the activity of ALK inhibitors that positively affect the prognosis of ALK+ NSCLC patients. The scientific content seems good as well as the English style and language used in the manuscript need. Moreover, I appreciate the scientific efforts to organize this paper and I think that the rationale is well-discussed. The resolution of the table of each section and the reference should be improved in order to meet the quality requirements of the journal. Moreover, each paragraph was clearly discussed and corroborated with what is exposed. I didn’t observe any remarkable incongruences throughout the text.

In my opinion, this article should be considered with minor revisions in light of comments and suggestions as indicated below.

1-      I would like to suggest the authors reformulate the abstract section in virtue of the fact that the key point of the review is focused on first-line randomized clinical trials of ALK-Is, mechanisms of resistance and future challenges in ALK+ NSCLC patients. From my point of view, it didn’t emerge at first glance the clinical significance of the review and I think that it would be more appropriate to highlight these clinical and pharmacological aspects. Please proofread this.

A1:  We thank the reviewer for this observation. We have reformulated the abstract as requested. Now we think that it is more appropriate at summarizing the contents of the review.

ALK translocation amounts to around 3-7% of all NSCLC. The clinical features of ALK+ NSCLC are adenocarcinoma histology, younger age, limited smoking history, and brain metastases are frequent sites of metastasizing.  The activity of chemotherapy and immunotherapy is modest in ALK+ disease. Several randomized trials have proven first that ALK inhibitors (ALK-Is) have higher efficacy respect to platinum-based chemotherapy and then that second/third generation ALK-Is are better than crizotinib in terms of median progression-free survival and brain metastases management. Unfortunately, most of the patients develop acquired resistance to ALK-Is mediated by on- and off-target mechanisms. This review focuses on the summary of first-line randomized clinical trials of several ALK-Is and the management of brain metastases with a focus on ALK-Is resistance mechanisms. The last section is addressed to future developments.

2-      The introduction paragraph should be more deeply investigated how molecular and biological characteristics of ALK rearrangements impact intracellular signaling pathways, briefly describing the role of involved partner genes and their deregulation effects. Please discuss it.

A2: We thank the reviewer for his suggestion. Now we have added in the introduction section a brief description of molecular pathways in which ALK plays a role.

Physiologically ALK is expressed only in the brain and spinal cord of embryos and it is essential for neurological development [Vernersson E, Khoo NKS, Henriksson ML, et al. Characterization of the expression of the ALK receptor tyrosine kinase in mice. Gene Expr. Patterns 2006; 6 (5), 448–461. https://doi.org/10.1016/j.modgep.2005.11.006]. In the adult life, ALK is constitutively expressed in limited nervous tissues. The aberrant expression and activation of ALK fusion proteins in cancer cells leads to cellular transformation through a signalling network which involves the activation of STAT3, AKT/PI3K, and RAS/ERK pathways, which are essential to rule cell proliferation, cell cycling, and survival [Huang H. Anaplastic Lymphoma Kinase (ALK) Receptor Tyrosine Kinase: A Catalytic Receptor with Many Faces.  Int J Mol Sci. 2018; 19(11): 3448. doi: 10.3390/ijms19113448]

3-      I suggest enlarging figure n.1 and improving its resolution. As it stands, it doesn’t seem sharp in line with the standard of the journal. The same should be applied to figure 2 with the difference that remains to specify the colour of each ALK domain in the figure caption. Please fix it.

A3: We accepted this point. Now both figures 1 and 2 are implemented according to the guidelines of the Journal.

4-      In the context of first-line randomized clinical trials of ALK-Is, I suppose that it is essential starting with an overview of first-line clinical trials and then list the cited studies. Please move the “Table 1 summarizes the global first-line clinical trials of ALK-Is” at the end of paragraph and insert the table there. I think that it's easier to follow and flow and it's less confusing. Please fix these suggestions.

A4: We confirm that we have placed the sentence at the end of the section.

5-      As described in the point 3, please fix the figure n.3 for a better resolution and indicate each reference in the caption. In the Table 4, the references are missing. Please report them (if available) for each study.

A5: Now we have implemented figure 3 according to the standards of the Journal. We have added the references related with table 4.  

6-      As well as for Figure 4 about the overview of resistance mechanisms to ALK inhibitors it should be improved both resolution and size.

A6: as above, we have augmented figure 4 according to the standards of the Journal.

7-       There are some mistakes in formatting table 5A and table 5B (correct form, not 5-A and 5-B). In particular, 5-B did not follow the guidelines of journal. Please adapt it in the same way as the previous ones.

A7: Now we have corrected the mistakes citing in the correct form Tables 5A and 5B and we have adapted table 5 according to the guidelines of the Journal.  

8-      Into the discussion section, I would like it if it motivates the choice of the optimal upfront ALK TKI for the first-line treatment of ALK+ NSCLC, the subsequent sequencing strategies, and whether these considerations should be based on specific on-target ALK resistance or not. And if indeed the goal of treating advanced ALK+ NSCLC should not just be limited to improve median PFS in the first-line setting. Please spend a few sentences on this point.

A8: We thank you the reviewer for his suggestion. Now we have added a brief sentence where we express the optimal upfront ALK-I for the first-line treatment and the subsequent strategies in the discussion section.

Both NCCN and ESMO guidelines have included alectinib, brigatinib or lorlatinib as the preferred option for the first-line treatment thanks to the great results attained in their pivotal randomized trials according to the long benefit on mPFS and optimal management of brain metastases which allow us to postpone brain radiotherapy in patients with asymptomatic and/or not large brain metastases. At disease progression, both the international guidelines recommend going on with the same ALK-I +/- local therapy in case of oligo-progression or in case of systemic progression to repeat a biopsy (recommendation but not mandatory) and choose the subsequent ALK-I on the basis of the acquired resistance mechanism. The optimal choice of upfront ALK-I and the subsequent sequency of other ALK-Is is not known due to the absence of prospective trials designed to investigate the optimal sequence of ALK-Is that could guarantee the best OS that should be our goal, while all the pivotal randomized trials have assumed as their principal endpoint the mPFS. Therefore, currently the optimal choice for the upfront ALK-I should be guided based on the available clinical data, patients’ features (e.g., with or without BM at the baseline) and the safety profile of the different ALK-Is which can affect the quality of life of the patients itself.  

9-      In the subparagraph about early-stage, we should need data to definitively answer the question of the utility of ALK-directed therapies in this setting. But I think that some considerations could be addressed in this context about the potential and limitations. Please add these suggestions to the discussion section.

A9: We accepted this point. Now we have added a sentence about these considerations in the subparagraph about early stage.

Both NCCN and ESMO guidelines did not recommend any ALK-I in the early-stage setting since we are waiting for the results from the trials of ALK-I designed and carried out in the perioperative stage. The introduction of the ALK-Is in this setting will depend on the results attained and if they will maintain the high premises derived from what they have produced in the metastatic setting. Clearly the follow-up of these trials must be longer to observe the effects on disease-free survival and above all overall survival that should be the principal endpoints to look at, since the early surrogate endpoints, such as major pathologic response or complete pathological response in the neoadjuvant trials are still not validated endpoints.     

10-   It is ascertained that after the failure of targeted therapies, chemotherapy remains a valid option, while the role of immunotherapy is yet to be clarified. Overcoming the challenges for the development of more potent drugs will be essential to improving the survival rate of ALK+ NSCLC in the future. What scenario could be expected? Please discuss rapidly in the conclusion.

A10: As requested also by reviewer 1, we have reformulated the conclusion section.

11-   Please recheck the reference list (they should be reported in this way: [1,2,3 etc] and not as indicated [1][2] throughout the text. Conflicts of Interest report some typing errors. Please correct this and fix some abbreviations and punctuation.

A11: we have corrected the reference list and typing errors. 

12-A good paper has to come from something that's inspired as well as a good song. So I agree with your acknowledgments to Lenny Kravitz, an incredible rockstar!

A12: thank you!